# Diagnostic and Prognostic Significance of Carboxypeptidase A4 (CPA4) in Breast Cancer

**DOI:** 10.3390/biom9030103

**Published:** 2019-03-14

**Authors:** Suleyman Bademler, Muhammed Zubeyr Ucuncu, Ceren Tilgen Yasasever, Murat Serilmez, Hakan Ertin, Hasan Karanlık

**Affiliations:** 1Department of Surgery, Institute of Oncology, Istanbul University, 34093 Istanbul, Turkey; sbademler@gmail.com (S.B.); hasankaranlik@yahoo.com (H.K.); 2Health Science Institute, Istanbul Gelisim University, 34310 Istanbul, Turkey; 3Department of Basic Oncology, Institute of Oncology, Istanbul University, 34093 Istanbul, Turkey; cerentilgen@gmail.com (C.T.Y.); serilmez55@hotmail.com (M.S.); 4Department of Medical Ethics and History, Istanbul University, 34093 Istanbul, Turkey; hakanertin@gmail.com

**Keywords:** breast cancer, CPA4, CPA4 mRNA, diagnostic analysis

## Abstract

Recent research focused on prolonged survival has suggested that carboxypeptidase A4 (CPA4) plays a role in both tumor microenvironment formation and distant metastasis in cancer. In some patients, serum and expression (mRNA) levels of CPA4 have been found to be correlated with the aggressiveness and progression of the disease. Accordingly, we conducted a first study to investigate the diagnostic and prognostic significance of CPA4 in the case of breast cancer (BC), the most common form of malignancy in women. The study included a total of 50 patients with BC and 20 healthy women as the control group. The participants’ serum CPA4 levels were determined by the ELISA test, and, for assessment of CPA4 mRNA, we used the PCR method. The serum CPA4 (*p* = 0.001) and CPA4 mRNA (*p* = 0.015) levels were found to be statistically significantly higher in the controls, compared to the patient group. When the results of patient group were statistically analyzed based on subgrouping by tumor characteristics, the measured CPA4 mRNA levels showed significant difference with respect to the molecular subtype (*p* = 0.006), pN status (*p* = 0.023), and pathological stage (*p* = 0.039), while the serum CPA4 measurements differed significantly in terms of pathological type only (*p* = 0.024). We conclude that CPA4 is diagnostically and prognostically not futile when used in combination with the other considerations and measurements in breast cancer.

## 1. Introduction

Breast cancer (BC) is the most common form of malignancy in women, accounting for 23% of all cancer cases and 14% of all cancer-related deaths. The annual incidence is around one million cases, of which some 200,000 (27% of all cancers in women) are reported in the US and 320,000 (31% of all female cancers) in Europe. Metastasizing in one-quarter of all patients, BC is a leading reason of death in women aged 30 to 59 years, particularly [1,2,3,4].

The primary risk factors include advanced age and a familial history of BC [5,6], and the accordingly implemented strategies of screening and early diagnosis have been associated with a reduced need for invasive intervention, increased overall survival, and decreased mortality [7,8]. The multiphase oncogenesis involves changes at the chromosomal level, polymorphism, angiogenesis, and uncontrolled cell growth that results in a tumor [9,10], with ongoing genetic mutations leading to invasion, metastasis, and resistance to treatment [11]. During all these phases, specific compounds are released from the cancerous cells in markedly increased amounts. Of these biomarkers, carboxypeptidase A4 (CPA4), a member of the zinc-containing family of metallocarboxypeptidases, is an exopeptidase that catalyzes the peptide bonds formed by the amino acids at the carboxyl end of the oligopeptides [12,13]. The molecular role of CPA4 is unclear. Cellular and biochemical characterization have shown that it is secreted from cells in proenzyme form (pro-CPA4). Once activated, it exhibits optimal activity at neutral pH, suggesting that it is likely to exhibit regulated activity within the extracellular environment. Potential biological substrates include peptides involved in proliferation and differentiation. Targeted activity of CPA4 against one or more substrates regulates the process of adipogenesis [14].

Various studies have shown that CPA4 is closely associated with the growth, differentiation, and aggressiveness of cancer cells. It has been suggested that CPA4, an important regulator of inflammation, also plays a role in tumor microenvironment (TME) formation and distant metastasis in cancer [12,13]. Subsequently, more studies conducted found marked increases in the CPA4 levels in patient tissue and serum samples and associated these findings with tumor progression and poor prognosis [15,16,17,18]. It was also reported that the high levels of CPA4 could be utilized as a biomarker of metastasis in certain cancers when assessed together with lymph-node involvement [19]. Although BC has been considered to be one of these pathologies, there has not been any current studies that examine the potential of serum CPA4 concentrations as a possible marker and prognosticator. In this study, we aimed to investigate the diagnostic and/or prognostic value of serum CPA4 levels in breast cancer. 

## 2. Materials and Methods

### 2.1. Data Collection

The study included 50 women who were followed up at the Istanbul University Institute of Oncology between 2017 and 2018 with a biochemical, radiological and pathological diagnosis of BC and 20 healthy adult controls. The control group consisted of women who had been visiting our breast surgery clinic for their routine check-ups, which had revealed no pathology. 

Serum samples were obtained from the participants upon admission, before any treatment was initiated. Blood samples were obtained by venipuncture and clotted at room temperature (RT). The serum samples were centrifuged (10 min, 4000 rpm) at RT and frozen immediately at −20°C until analysis.

This study was approved by the Istanbul Faculty of Medicine Ethics Committee (2016/1216). Based on the following additional ethics consultation, the study was designed and conducted in accordance with the principles of the Declaration of Helsinki. Written informed consent was obtained from all participants. 

### 2.2. Methods Used

#### 2.2.1. Determination of the Serum CPA4 Levels by Use of the ELISA Test

Forty µL of serum sample and 10 µL of CPA4 antibody were placed in the antibody-coated wells by using an automatic pipette while a volume of 50 µL of standard preparation was placed in the other wells. The wells were incubated for 45 min at 37 °C. To each well was then added 50 µL of streptavidin-HRP. For the formation of the antigen–antibody complex, the wells were incubated again for 30 min at 37 °C. After washing five times with 300 µL washing liquid and thorough drying, a volume of 50 µL of chromogen solution was added before incubation for 15 min at 37 °C. The color reaction was stopped by adding 50 µL of stopping solution. The absorbance and concentration values of the samples were measured respectively by the ELISA reader (ChroMate 4300 Microplate Reader, Palm City, FL, USA) at 450 nm. The concentrations were calculated, based on the curve plotted according to the standards with known values were compared with the concentrations determined automatically. 

#### 2.2.2. Determination of the Gene Expression Levels

##### RNA Isolation

A total of 200 µL of serum and 800 µL TRIzol were homogenized in 1.5-ml Eppendorf tubes, to be followed by 5-min incubation at RT. After adding 200 µL of chloroform and then another 5-min incubation at RT, the product was centrifuged for 15 min at 12,000 xg. The supernatant out of the phased homogenate was transferred into fresh 1.5-ml Eppendorf tubes. After adding 550 µL of isopropanol, the product was incubated for 5 to 10 min at RT and then centrifuged for 10 min at 12000 xg. The supernatant was removed and, after adding 1 ml of 75% cold ethanol, the pellet was vortexed and centrifuged for 5 min at 7500 xg. The supernatant was removed, and the resultant RNA was dissolved in RNase-free water. With this method, RNA is separated from DNA after extraction with acetic acid solution and chloroform, followed by centrifugation. Under acidic conditions, total RNA remains in the upper aqueous phase, while most of DNA and proteins remain either in the interphase or in the lower organic phase. Total RNA is then recovered by precipitation with isopropanol and can be used for several applications. 

##### Complementary DNA (cDNA) Synthesis

cDNA synthesis from total RNA was carried out by use of a commercially available kit. Six microliters of total RNA, 0.5 µL of random hexamer primer, and 6 µL of dH_2_O were placed in 0.5-ml PCR tubes. The reagents 10 mM deoxynucleotide mix (1 µL), DTT stock solution (1 µL), RNase inhibitor (1 µL), and SCRIPT reverse transcriptase (1 µL) were then mixed, respectively, and placed in a conventional PCR apparatus. The synthesis conditions are shown in Table 1. The resultant cDNA samples were stored at –20 °C. 

Six microliters of total RNA were used in real-time PCR for CPA4 gene expression. The primers TGCAACACAATGAAGGGCAAG and CGGCAATGTTGTCCATCTCG were used as the forward and reverse primers, respectively. 

In the analysis of the CPA4 expression levels, RT-PCR was performed with the Light-Cycler 480 (Roche Penzberg, Germany) using the SYBR GreenMaster PCR Kit (Jena Bioscience, Erfurt, Germany). GAPDH was used for mRNA expression normalization. The RT-PCR conditions were set according to the GreenMaster PCR protocol. 

Melting curve analysis was then performed to confirm the RT-PCR products and exclude the non-specific products and by-products, such as primer dimers. The analysis is based on the melting temperatures of the products, heated from 55 °C to 95 °C at a pace of 0.2 °C/sec. Depending on the product structure, the melting curves are obtained at approximately 75 °C to 80 °C.

In RT-PCR, the cycle where the level of fluorescence exceeds the measureable threshold is referred to as the threshold cycle (Ct). We performed our assessment of gene expression by using the 2-ΔΔCt method based on the Ct values obtained. 

#### 2.2.3. Statistical Analysis

Variables were investigated for normal distribution graphically (by histograms and possibility charts) and analytically (by the Kolmogorov–Smirnov/Shapiro–Wilk tests), while the Mann–Whitney U test was used for non-normally distributed variables. Receiver-operating characteristics (ROC) curve analysis was performed to determine the diagnostically optimal cut-off values of serum CPA4 and CPA4 mRNA. For each variable, the score with the maximum sensitivity and specificity was selected as the approximate cut-off value. The area under the curve was also calculated. All statistical analyses were performed by using SPSS v. 21 (IBM, Armonk, NY). *p* < 0.05 was considered statistically significant.

## 3. Results

There was no statistically significant difference between the patient and control groups in terms of mean age (*p* = 0.496). The levels of serum CPA4 (*p* = 0.001) and CPA4 mRNA (*p* = 0.015) were found to be higher in the control group, compared to the BC group (Table 2). 

When the results of the BC group were statistically analyzed based on subgrouping by tumor characteristics, the measured CPA4 mRNA (gene expression) levels showed significant difference, with respect to the molecular subtype (*p* = 0.006), pN status (*p* = 0.023), and pathological stage (*p* = 0.039), while the serum CPA4 measurements differed significantly in terms of pathological type only (*p* = 0.024) (Table 3).

The CPA4 cut-off value for the ELISA test was 7.5, at which the test showed 70% sensitivity and 72% specificity. The CPA4 cut-off value for the PCR method was found to be 3.54, at which 75% sensitivity and 72% specificity were observed (Table 4). We found no statistically significant correlation between the CPA4 ELISA and PCR methods (*p* = 0.690) (Figure 1).

## 4. Discussion

The mortality of BC can be as high as 25% in case of metastasis [1,2]. Although many risk factors have been identified for BC [5,6], early diagnosis can reduce the need for invasive intervention, increase the overall survival, and decrease the mortality [7,8], and can be achieved by laboratory detection of certain markers produced during the growth and proliferation of the cancerous cells. It is thought that CPA4, one of these markers common to different tumor types, is effective in the TME formation during oncogenesis [11,12,13]. It has been suggested that the presence of high CPA4 expression can be used as a precise marker of metastasis in certain cancers [14,15]. However, the literature appears to contain no study addressing the diagnostic value of CPA4 in BC, by measuring both expression and serum levels [14]. Accordingly, our study is a first study using the two different measurements of CPA4 in assessing the potential use of the molecule as a marker in BC. 

There have been several studies indicating that CPA4 might be of clinical benefit in the early diagnosis of certain cancers. In one of these, Sun et al. reported markedly increased CPA4 levels in the tumor tissues and serum samples of patients with pancreatic cancer and associated this marker with tumor progression and poor prognosis [16]. Another study by Sun et al. argued that the CPA4 levels measured in patients with liver cancer were closely correlated with hepatocarcinogenesis and that high CPA4 expression might be a sign of poor prognosis [17]. The following studies found an excessive expression of CPA4 in soft tissue and non-small-cell lung cancers and reported that it could be used as an early sign of poor prognosis. However, in patients with non-small-cell lung cancer, it has also been recommended that the CYFRA21-1 levels should be considered besides the CPA4 levels [18,19]. Accordingly, it is arguable that adequate prognostication by serum CPA4 levels depends on the specific type of cancer in question. In our study, serum CPA4 and CPA4 gene expression levels were found to be significantly lower in the patients with BC, compared to the controls (*p* < 0.05). We therefore conclude that measured levels of serum CPA4 and CPA4 mRNA per se cannot be used as a marker in the early diagnosis of BC. CPA4 may act as a negative regulator of adipogenesis and downregulation of CPA4 may represent an integral part of the adipogenic program as well as the pro-adipogenic effects of FGF-1 [20]. 

There are a variety of techniques to assess the CPA4 expression in a tumor [21]. To determine the CPA4 levels in our study, we used two different techniques based on measurement of CPA4 gene expression and CPA4 (ng/L) protein level. The fact that the sensitivity and specificity values were, respectively, 70% and 72% for the CPA4 ELISA test and 75% and 72% for the CPA4 PCR test signifies a moderate-to-good test reliability. We also compared the results of both tests based on tumor characteristics. The measured CPA4 mRNA levels showed significant difference with respect to the molecular subtype (*p* = 0.006), pN status (*p* = 0.023), and pathological stage (*p* = 0.039), while the serum CPA4 measurements differed significantly in terms of pathological type only (*p* = 0.024).

## 5. Conclusions

We conclude that CPA4 is prognostically and diagnostically not futile in breast cancer, when used in combination with the other considerations and measurements. Despite our primary limitation of low-participant number and our contradicting results, we think that our study is the first study examining the potential of serum CPA4 concentrations as a possible marker and prognosticator and is preliminary to further the studies to more extensively address the association between the molecule and the known clinical parameters in BC. 

## Figures and Tables

**Figure 1 biomolecules-09-00103-f001:**
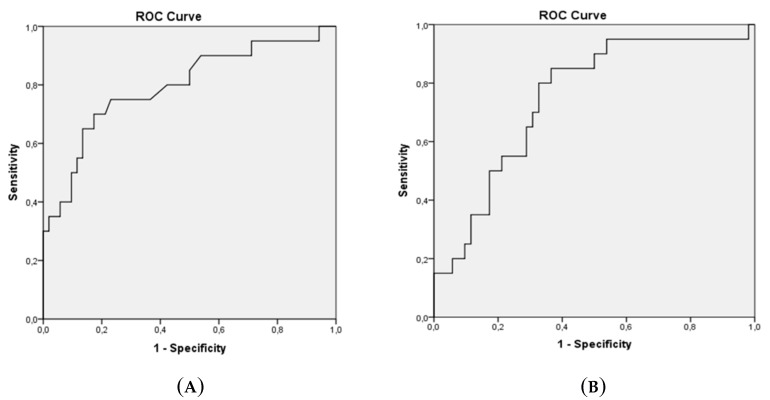
*CPA4* cut-off values for the ELISA (**A**) and PCR (**B**) methods.

**Table 1 biomolecules-09-00103-t001:** The cDNA synthesis conditions.

Temperature (°C)	Duration (min)
42	10
50	60
70	10

**Table 2 biomolecules-09-00103-t002:** Comparison of the two groups.

	BC Group (*n* = 50)	Control Group (*n* = 20)	*p*
Age	50.96 ± 11.86	48.5 ± 11.32	0.496
Level of serum *CPA4* (ng/L)	8.04 ± 10.10	28.67 ± 35.93	<0.001
Level of *CPA4 mRNA*	6.80 ± 13.82	83.32 ± 217.02	0.015

**Table 3 biomolecules-09-00103-t003:** The patients’ serum *CPA4* and *CPA4 mRNA* levels as analyzed based on subgrouping by tumor characteristics.

	*CPA4* (ng/L) Protein Level	*p*	*CPA4* Gene Expression	*p*
**Tumor diameter**		0.462		0.632
≤2 cm (*n* = 18)	5.07 ± 3.24		5.62 ± 13.29	
>2 cm (*n* = 32)	9.08 ± 11.45		7.21 ± 14.16	
**pN status**		0.160		**0.023**
pN0 (*n* = 22)	10.31 ± 13.59		11.75 ± 19.20	
pN-positive (*n*=28)	6.25 ± 5.82		2.91 ± 4.93	
**Pathological type**		**0.024**		0.472
Invasive ductal carcinoma (*n* = 38)	6.27 ± 5.63		6.07 ± 13.44	
Invasive lobular carcinoma (*n* = 4)	19.95 ± 28.14		3.16 ± 2.93	
Other (*n* = 8)	10.47 ± 10.13		12.06 ± 18.34	
**Lymphovascular invasion**				
LVI-negative	10.70 ± 14.28		8.60 ± 16.07	
LVI-positive	5.92 ± 4.67	0.183	5.73 ± 13.90	0.558
**Molecular subtype**				
Luminal (*n* = 36)	8.95 ± 11.08	0.309	3.51 ± 8.74	**0.006**
Non-luminal (*n* = 14)	5.68 ± 6.76		15.24 ± 20.19	
**Grade**		0.558		0.073
1 (*n* = 5)	9.02 ± 9.53		3.19 ± 5.22	
2 (*n* = 17)	10.19 ± 14.99		1.40 ± 2.15	
3 (*n* = 28)	6.58 ± 6.70		10.84 ± 17.38	
**Pathological stage**		0.317		**0.039**
1A (*n* = 12)	5.38 ± 3.18		6.08 ± 13.07	
2A (*n* = 10)	13.58 ± 18.67		18.46 ± 23.19	
2B (*n* = 21)	8.14 ± 7.82		2.77 ± 4.56	
3A (*n* = 3)	3.66 ± 3.45		0.01 ± 0.10	
4 (*n* = 4)	7.90 ± 2.24		6.05 ± 7.53	

**Table 4 biomolecules-09-00103-t004:** Sensitivity and specificity at the cut-off points.

	AUC	*p*	Sensitivity	Specificity	Cut-Off Value
*CPA4* ELISA	76.6	0.001	70%	72%	7.5
*CPA4* PCR	76.4	0.001	75%	72%	3.54

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
