# Peer review of "Diagnostic and Prognostic Significance of Carboxypeptidase A4 (CPA4) in Breast Cancer"

_biomolecules, 2019, doi:10.3390/biom9030103_

Round 1

Reviewer 1 Report

Reviewer comments on manuscript from Bademler et al., ID : biomolecules-451980

The manuscript from Bademler et al. describes the evaluation of serum CPA4 mRNA and protein level as a diagnostic / prognostic marker in breast cancer. Some English changes, methodological and statistical concerns should be addressed before this short manuscript would be ready for publication in Biomolecules.

English improvement and spell check:

               - Many words are not properly cut in the printable version. For example : lines  43, 44, 45, 46, 51, 135… Please check the whole manuscript.

               -  In the method section, µL become « @L Â» ??

               - English language of sentences from lines 53 to 60, line 126 and in line 157 should be improved.

               - Discussion line 148 = introduction line 36.

Methods :

               - No ethic committee approval is reported for data collection

               - The cell origin of RNAs extracted from sera for CPA4 assessment is not clear : white blood cells mixed with cancer prone vesicles ? What do the authors expect ?

               - RNA isolation with Trizol@, reverse transcription and PCR protocols are sometimes described with too many details (citing the commercially available kits seems sufficient) but some information are still missing : the authors indicate that the RT is performed with 6µL of total RNA ; what does it mean in terms of quantity ? Please provide the sequence of the primers used for real- time PCR.

Results

               - line 127 : levels of serum CPA4(p=0.000) ???

               - The major concern of this study is the intra- group variability of CPA4 protein or mRNA levels which are yet interpreted as statistically reliable by the authors for tumor diagnosis and pathological stage prognosis. Indeed, the authors claim that CPA4 protein (or RNA) level is statistically higher in control group than in breast cancer patient group but the data are for example 8±10ng/L vs 28±35ng/L…how can a protein level be a negative concentration ? The same concern exists for the data from table 3 (pN status, pathological stage… Therefore, the statistics  (for example P<0.001) sound questionable. Could the authors provide details on statistical methods and box plots of CPA protein /mRNA levels for each tumor characteristic ?

                - Could the authors provide detailed explanations about figure 1: how is the cut-off determined for example ?.

Author Response

first of all thank you for your time. 

We rewrite our manuscript according to your suggestions. We add our report at the and as a word file. 

Reviewer 2 Report

The authors investigated the diagnostic and prognostic potential of serum levels and mRNA expression of CPA4 in 50 breast cancer patients and 20 controls. Levels of both serum and mRNA expression were lower in breast cancer patients as compared to controls. However, the analyses supporting this conclusion are limited and the number of included samples is very small.

Major comments.

1. Patient characteristics should include information on hormone-receptor status.

2. Are clinical variables including age, gender, year of sample collection, etc. matched to those of patients?

3. The SD value of CPA4 levels (both serum and expression) is very large in the controls group. A plot (e.g. scatter plot) should be shown to demonstrate how many of the control samples had same levels of CP4 as BC samples.

4. CPA4 mRNA levels are associated with molecular subtype, pN status, stage and serum levels with type. Variables with significant association should be included in multivariate analysis, which are missing.

5. The cut-off value for CPA4 serum/gene expression should be established based on its expression in the control group. It is not clear from the manuscript how the value was established.

6. Authors report no significant correlation between mRNA and protein levels, however the Pearson’s R value should still be included to give readers an idea of what the correlation is. Was the correlation different in BC samples vs. controls? A plot showing the correlation would be beneficial.

7. Is any survival data available for this patient cohort? Additional survival analysis (Kaplan-Maier plots) would provide stronger evidence for the lack of prognostic value. If these data is not available, authors should look at validating their findings in public datasets e.g. TCGA or km-plotter (PMID:20020197).

Author Response

First of all thank you for your time

We rewrite our manuscript according to your suggestions. We add a word file at the end. 

Round 2

Reviewer 1 Report

The boxplots I requested were not provided but but the manuscript has been globally improved and may be suitable for publication.

Author Response

We put box plot but I think they are small and understandable.

Reviewer 2 Report

The authors have addressed all questions. Figures provided in the response could be moved to the main manuscript for clarity.

Author Response

thank you for your time